# Sexual orientation disclosure and depression among Thai gay, bisexual, and other men who have sex with men: The roles of social support and intimate partner violence

Eduardo Encina[1,2], Worawalan Waratworawan[1,3], Yamol Kongjareon[1], Mayur M. Desai[2], Thomas E. Guadamuz[1,4]*

1 Mahidol Center for Health, Behavior and Society, Faculty of Tropical Medicine, Mahidol University, Bangkok, Thailand, 2 Department of Epidemiology of Microbial Diseases, Yale School of Public Health, New Haven, CT, United States of America, 3 Department of Society and Health, Faculty of Social Sciences and Humanities, Mahidol University, Nakhon Pathom, Thailand, 4 John F. Kennedy School of Government, Harvard University, Cambridge, MA, United States of America

* tguadamu@hotmail.com

**Data Availability Statement:** Data are available upon request from the Center of Excellence in Research on Gender, Sexuality and Health

## Abstract

### Background

Among gay, bisexual, and other men who have sex with men (GBM), sexual orientation disclosure to social groups can act as a significant risk for depression. The primary goal of this research is to understand the association between disclosure and depression, the association of social support and intimate partner violence (IPV) experiences, depression, and disclosure.

### Methods

This project uses a secondary dataset of Thailand from a larger cross-sectional study distributed in the Greater Mekong Sub-Region. This study utilized web-based answers from 1468 Thai GBM respondents between the ages of 15–24 years.

### Results

Prevalence of depression was over 50%. Across the social groups of interest, those who disclosed to everyone had the lowest depression prevalence. This association was statistically significant for all groups ($p<0.050$) except for "Family members" ($p = 0.052$). There was a statistically significant association illustrated between full disclosure to social groups and increased social support. Most respondents (43.9%) had low social support, and additionally this group had the highest level of depression, compared to those with high social support. There was a statistically significant association for lowered depression outcomes and increased social support. IPV experiences that occurred within the last six months had a statistically significant relationship with depression ($p = 0.002$). There was a notable association between those with experiences of being a victim of IPV, alone and in conjunction with experience of being a perpetrator of IPV, which was associated with increased odds of

(MUGSH), Mahidol University 999 Puttamonthon 4 Road Salaya Nakhon Pathom, Thailand 73170, or by emailing Ms. Mudjalin Cholratana at mudjalin@gmail.com. Data cannot be deposited into a public repository because of ethical restrictions. While the data have been stripped of all identifiers, there exist potential risks where sensitive data on substance use and sexual behaviors may be traced back to specific participants. For this reason, MUGSH can be contacted on a case by case basis to retrieve data from this study.

**Funding:** Materials for this research project was funded by Mahidol University. Eduardo Encina was supported by the Yale School of Public Health Summer Fellowship Award 2020. Thomas E. Guadamuz, Worawalan Waratworawan, and Yamol Kongjareon were supported through NIAID grant R21AI140939 and NIMH grant R01MH119015. The funders had no role in study design, data collection and analysis, decision to publish, or preparation of the manuscript.

**Competing interests:** The authors have declared that no competing interests exist.

depression. However, the type of IPV experiences an individual had did not differ based on disclosure status.

## Discussion

This study provides strengthened evidence of the impact that differences in supportive networks can have on mental health outcomes. In addition, they provided a wider consideration for how people may have different IPV experiences, either as a perpetrator, victim, or both, and how those shapes health outcomes of depression. GBM communities still face adversity and challenges that affect their long-term health outcomes, even if they do live in what is considered an accepting country.

## Introduction

One of the most important expressive processes among lesbian, gay, bisexual, transgender, queer, and questioning (LGBTQ+) individuals is "coming out", wherein individuals navigate their sexual identity and whether to disclose it to others [1, 2]. A few of the influential environmental and social factors that can impact when and to whom an individual discloses their sexual orientation include items such as self-esteem, potential for acceptance, perceived level of emotional or physical harm, and social stigma [3]. One potential buffer between the stress of coming out and worsened mental health outcomes is having a community that is supportive of and empowering towards LGBTQ+ individuals [4–6]. Without such a community, there can long-term, negative health impacts along with a decreased sense of self-comfort, acceptance, and identity development [7]. A family is one social group that can influence personal development and well-being, shown by studies supporting an association between family rejection related to sexual orientation and increased negative health outcomes [3, 8, 9].

### Sexual orientation disclosure

While processing the decision of "coming out" to different social networks, gay and bisexual men grapple with the potential responses from others, including positives ones of support and acceptance, or negative ones of social disapproval, bodily harm, and avoidance [10–12]. This can cultivate a variety of stressors that over time can translate into self-hatred, isolation, and increased violent behaviors [10–12]. These negative stressors related to coming out can even compound by one's race and ethnicity, which can further one's risk for depression [2, 12]. The minority stress theory is a conceptual framework developed to describe the numerous stressors that may uniquely face members of minority groups [13]. It describes individual and external stressors, either psychological, physical, or social, that can afflict members of stigmatized social groups, and which may accrue with time, leading to health deficiencies [13, 14]. This theory encapsulates distal and proximal stressors that GBM communities may experience, ranging from the distal stressors of antigay violence, and homophobia to proximal stressors of internalized homophobia, along with increased concealment out of fear of rejection [13, 15].

These stressors can multiply among those who hold multiple marginalized identities in their respective communities, such as GBM who also identify with a marginalized racial/ethnic background, and negative health outcomes [14, 15]. Previous research has reported that for communities who experience minority stress, there have been positive associations illustrated between internalized homonegativity and identity concealment with same-sex partner

violence perpetration, while increased community support can make people less vulnerable to such stressful experiences [15, 16]. Previous studies have analyzed the associations of perceived social support and mental health-related functioning using a Social Provisions Scale (SPS), a validated measure to study perceptions of social support in one's environment, making it an adept variable to analyze mental health associations among Thai GBM communities [17–19]. Social support can play a vital role in supporting human health, as direct links have been shown between social support and increased mental health outcomes, such as increased self-esteem and longevity [20]. In fact, previous research has attempted to understand the nuances with social support constructs by analyzing the variations in support from family and friends. Generally, studies have exhibited that family support can be a protective agent against negative mental health outcomes and when this family support is lacking, friend social support can be just as impactful [20, 21]. Research have shown that white respondents experience the lowest levels of stigma, followed by Hispanic/Latino, then Asian respondents, and then Black respondents having experienced the highest levels of stigma [15]. Social stigma towards gay and bisexual men within Western cultures is well-studied, however it may differ in subtle manners, compared to Southeast-Asian countries.

Thailand sentiment towards those who identify as LGBTQ+ has been popularized in urban areas through descriptions of being a "gay paradise" or a "safe haven [22]." Despite increased liberal attitudes and protective Thailand public policy, LGBTQ+ individuals have shown preference towards presenting as heterosexual and following heteronormative gender roles to actively avoid gossips, anti-gay comments, and ensure their own job security [23]. This heteronormative pressure can impact how people disclose their sexual orientation and with whom. Research has suggested that men are more likely to experience sexual violence perpetration and victimization if their social networks are comprised of more "closeted" gay friends or strictly sex partners [24]. Without a supportive environment comprised of individuals who gay and bisexual men trust and who they share positive character traits with, men may have trouble processing emotional and physical discrimination that can fester feelings of depression and internalization homonegativity, factors which have been associated with IPV [16, 24]. When considering reports that almost all MSM had experienced some form of homophobia in one study, and another study which reported that half of MSM discrimination events are connected to sexual orientation or gender identity, this becomes especially relevant [25, 26]. There has been an increase in recent years towards research catered to Thai gay and bisexual male communities, however there still remains a gap in understanding the effect that sexual orientation disclosure has on depression and the potentially influential role of other variables, including social support and IPV.

## Intimate partner violence

IPV occurs when intimate partners or spouses engage in interpersonal violence, which can transpire as either physical, psychological, financial, or sexual [27]. While well-documented among women in heterosexual relationships (15–71%), IPV experiences remain scarcely studied among same-sex relationships [28]. Estimates from several studies analyzing IPV prevalence among LGTBQ+ individuals range from 5.75% to 54% for MSM [29–32]. Deleterious health outcomes can result from experiencing some form of IPV, either physical (e.g. injuries, HIV infection) and/or psychological (e.g. chronic mental illness, depression) in nature [33]. The impacts from having a history of IPV, such as verbal abuse or forced sex, can be multi-layered, leading to lowered incidence of healthy self-image, an increased predisposition to depression, and sexually transmitted infections [34, 35]. A study elucidating depression risk factors among Canadian gay and bisexual men discovered most respondents having experienced at

least some form of anti-gay violence (i.e. bullying, harassment, and physical violence) [36]. Considering ones potential feelings of self-insecurity, the homophobic attitudes of ones surrounding community, and the minimal availability of accessible and healthy emotional outlets, this can further ones susceptibility to long-term depression outcomes and IPV. IPV experiences are potentially thought to have a higher impact on health, including comorbidities among same-sex couples because of elevated substance/alcohol use, the scarcity of health resources commonly seen in same-sex couplings, and social stigma [35].

The majority of IPV research does not differentiate between "victims" or "perpetrators", an important distinction as there may be unique risk factors among consistent IPV perpetrators such as history of trauma or internalized self-esteem issues. The lack of research is concerning, especially considering that 18.4% of MSM respondents in one study reported a history of forced sex, and of this, a majority of 67.3% reported forced sex on more than one occasion [37].

## Current study

The present study aims to examine the associations between sexual orientation disclosure among Thai GBM to their social groups and depressive outcomes. Additionally, this study aims to understand the extent that social support and IPV may mediate the relationship between depression and full disclosure. This study is focused on using this knowledge to inform future policy and enhance community efforts to provide resources for Thai GBM who may experience negative emotional health outcomes related to either violent behaviors and sexual orientation.

## Methods

### Sample

Data for this study comes from an online, multi-country cross-sectional project, "Greater Mekong–Young MSM Internet Survey", aimed at describing HIV risk behavioral trends, prevalence, and syndemics factors. Data collection occurred in ten countries throughout the Greater Mekong sub-region (i.e. Cambodia, Hong Kong, Indonesia, Lao PDR, Malaysia, Myanmar, Philippines, Singapore, Thailand, Vietnam). The study was a collaboration between Mahidol University and the Asia Pacific Coalition on Male Sexual Health (APCOM).

Data was collected between 30 March and 31 May 2018. Recruitment occurred through a web-based behavioral and epidemiological survey which was disseminated through several popular social networking websites and gay-oriented dating applications, as well as through NGO partner provided email lists with potential participants. Interested participants who accessed the survey entered a welcome page, then were provided with a unique ID and preferred language for the survey. The respondent's eligibility was then assessed, and upon acceptance, they were given a study overview and information to contact the study staff.

Mahidol University Social Sciences Institutional Review Board (MU-SSIRB) reviewed and approved the study (Certificate of Approval No. 2016/040.0902). For participants younger than 18 years, MU-SSIRB approved waivers of parental permission, informed by our previous study on proxy permission and waivers of parental permission [38]. All participants reviewed a participant information sheet at the first page of the online survey and then clicking through the assent/consent process. Details of the study's risks and benefits and contact information for the MUSSIRB chairperson were included in the consent process. Participation in this study was voluntary and anonymous. The secondary data analysis was reviewed by Yale University Institutional Review Board and was deemed exempt from IRB review and approval.

## Measures

**Depression.**     To accurately assess participant mental health, the survey included the Center for Epidemiologic Studies Depression Scaled Revised (CESD-R-10), a self-reported depression measure—shortened version of the original Center for Epidemiologic Studies Depression Scale (CES-D) [39, 40]. The CESD-R-10 scale is comprised of various items probing into commonly associated symptoms and behavioral patterns of depression. The CESD-R-10 has been well-used within previous epidemiological studies and is considered a valid and accurate tool for discerning or predicting emotions and behaviors that algin with depressive outcomes [41–43]. A participant responses to questions on the CESD-R-10 are scored on a scale of 0–3. A respondent is considered missing if they fail to answer more than two items on the survey, but otherwise their total score is calculated by summing the remaining 10 items. A dichotomized variable was created based on this scale. A participant was considered depressed and coded "1" if they scored 10 or above, but if they scored below 10 then they were not considered depressed and were coded "0".

**Sexual orientation disclosure.**     There were four defined social networks that participants were asked to estimate the degree to which they had disclosed their sexual orientation to the people in those groups. The first three networks comprised friends or colleagues through: (1) "Other people in the same school/university/workplace", (2) "Friends outside school/university/workplace", and (3) "Teachers in your school/university/boss in your workplace", which are subsequently referred to as "People in school/work", "Friends outside school/work", and "Teachers or Boss". The fourth social network was participants' (4) Family members. Among each of the networks, an individual's response was coded as "1" if they had told everyone, or their response was coded as "0" if they stated that they had not told everyone.

**Social support.**     The "Scale of Social Provisions-10 items (ÉPS-10)" is a shortened scale that participants were presented with, and based on their responses, it was used to build a continuous variable of social support. The ÉPS-10 has been validated in multiple studies and is comprised of questions analyzing attachment and social integration with scores attached to each answer, in order to gauge social support availability [17–19, 44]. The scale has a total possible score out of 40, and participants are considered to have higher social provisions with a higher score and lower social provisions with lower scores. There are no defined thresholds that distinguish between low or high social support with the ÉPS-10, and so this study set them on the 33rd (30), 66th (34) and 99th (40) percentile of respondent scores to create a 3-level categorical variable (1 = Low support, 2 = Medium support, 3 = High support).

**Intimate partner violence.**     Four questions were related to IPV experiences within the last 6 months (i.e. "Hurting, hitting, slapping the body of a regular partner, casual partner or male sex work partner", "Forcing a regular partner or casual partner to have sex", "Forcing a person who is a male sex work partner to have sex." and "Fondling or unwanted sexual touching."). For each question, participants explicitly answered whether they had never experienced the behavior described, and if they had experienced it, they identified whether they were the perpetrator, the victim, or both. Participants who responded that they "Never" engaged in any IPV experiences were coded as "0". Those who were identified as an IPV perpetrator were coded as "1". Participants could have answered "Never" or "I did it" to any of the four defined behaviors for this label, but they had to have answered "I did it" for at least one of the questions. Those who were identified as a victim of IPV were coded as "2". Participants could have answered "Never" or "It was done to me", but to be classified as an IPV victim they must have answered "It was done to me" to at least one of the four defined behaviors. Participants who were identified as both a perpetrator and victim of IPV were be coded as "3". In order to be classified as a dual perpetrator and victim of IPV, participants could have answered a

combination of "Never", "I did it", "It was done to me", or "Both" to any of the four behaviors, but they must have answered in a manner that indicated having performed both actions.

**Demographics and social characteristics.** Demographic and social characteristics of area of residence, age, sexual orientation, work, and education status, along with income per month, and religion were acquired from each participant. The "Age" variable was dichotomized (0 = 15–17, 1 = 18–24) and distinguished between teen and adult participants. The variable that defined a person's highest achieved education was classified from the original six categories to three (1 = Secondary or less, 2 = Vocational, 3 = Bachelor's or higher).

## Statistical analysis

The sociodemographic characteristics of the sample were summarized using descriptive statistics and were followed by chi-square tests to examine the associations between the sociodemographic variables with depression prevalence among participants. Both chi-square tests and logistic regression modeling were used to examine the distribution of the four sexual orientation disclosure variables, IPV experiences, social support, and their respective associations with depression. Sociodemographic characteristics were adjusted using multivariable models. We examined the associations of each disclosure variable with social support levels and IPV experiences to discern whether social support and IPV may be statistical mediators of the association between disclosure and depression. Finally, a series of three logistic regression models were run for the four sexual orientation disclosure variables to test the hypothesized mediating effect. These three logistic regression models, in addition to the disclosure and sociodemographic control variables, included 1) level of social support, 2) experience of IPV, and 3) both potential statistical mediators. We identified the attenuation of the effect of disclosure on depression outcomes with the addition of these potential statistical mediators to be supporting evidence for our hypothesized mediation pathway. Any $p<0.05$ was defined as statistically significant. The statistical analysis software SAS, version 9.4 [SAS Inc., Cary, North Carolina], was used to perform all analyses. The aim of this study was to understand the relationship between sexual orientation disclosure on an individual's mental health and the mediating influence of possibly related variables (i.e. social support and IPV experiences). To properly understand any association between sexual orientation disclosure and the potential mediating effects of social support and IPV, several associations were probed. Fig 1 illustrates the identified six pathways of interest.

## Results

### Descriptive statistics

As shown in Table 1, the sample consisted of 1468 total respondents who answered all survey questions and were included in the final analysis. Adults were the primary respondents (1287; 87.7%), of whom most (586; 39.9%) lived in the capital city of Bangkok, followed closely by those (531; 36.2%) who lived in a different Thai city. Among total respondents, most self-identified as "Gay" (1209;82.4%) and to a lesser extent, 217 (14.8%) self-identified as "Bisexual". Many of the participants held a degree of religious value in their lives, specifically finding it to be very important (712; 48.5%) or somewhat important (616; 42.0%). A third of respondents consistently disclosed their sexual orientation to everyone a part of all their social groups, with individual disclosure to "All" highest among "Family members" (37.4%) and lowest among "Friends outside school/work" (28.1%).

A common trend of higher prevalence of depression was identified in each of the four social groups of interests for those who had "Not [told] everyone", specifically with the highest prevalence among those who did not disclose their orientation to their "Teachers or Boss" (59.7%).

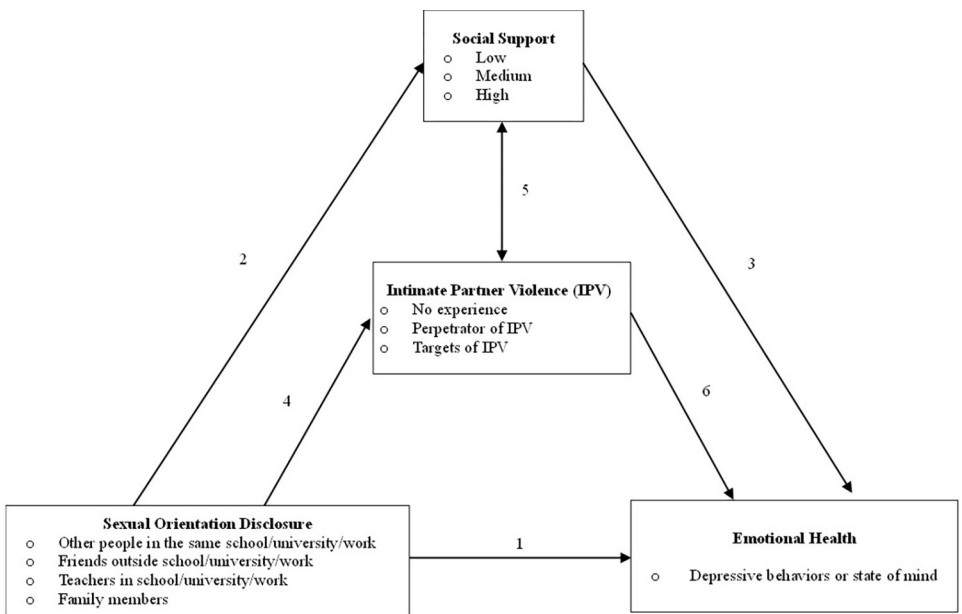

**Fig 1. Hypothesized model of the impacts that sexual orientation disclosure has on emotional health outcomes.** *1* –sexual orientation disclosure direct impact on emotional health. *2* –the effect of sexual orientation disclosure on levels of social support. *3* –social support effects on emotional health. *4* –sexual orientation disclosure effect on IPV experiences. *5* –the total association between individual social support and IPV experiences. *6* –IPV experiences association with emotional health outcomes.

Except for the social group of "Family members" ($p = 0.052$), every other group had a statistically significant association between full disclosure and lower prevalence of depression ($p<0.050$). Among the majority who were categorized with low social support (43.9%), the highest depression prevalence (65.1%) was identified while among those who had high social support had a lower identified depression prevalence (46.4%). There were no IPV experiences reported among most respondents (71.9%), but among the 108 (7.4%) who were identified as IPV "Victims" was the highest prevalence of depression (69.4%). This contrasts with the lower prevalence of depression (53.6%) among those with no IPV experience. The association between social support and depression was seen to be statistically significant ($p<0.001$) along with types of IPV experiences and depression ($p = 0.002$).

## Mediational analysis

In the unadjusted and adjusted analyses presented within Table 2, "Friends outside school/work" (OR 0.64; 95% CI 0.51–0.81) and "Teachers or Boss" (OR 0.63; 95% CI 0.50–0.79) both had the lowest odds of depression based on their full disclosure behavior. Overall, our analyses revealed a trend of higher social support along with lower odds of depression. There was no apparent statistically significant association among IPV perpetrators (OR 1.05; 95% CI 0.67–1.65), however depression among IPV victims (OR 1.97; 95% CI 1.28–3.02) was statistically significant. Adjustment for sociodemographic variables did not significantly alter the ORs (95% CI) for full disclosure to any of the social groups, levels of social support, or IPV Types.

Table 3 describes the analysis looking at the associations of sexual orientation disclosure with IPV type and social support. Regardless of the social group, the majority of those who were categorized with low social support did not disclose to everyone, however with higher levels of support, there was a steady increase across all groups in the percentage of those who

**Table 1. Sociodemographic characteristics among GBM study sample (*n* = 1468) and their associations with depression.**

| Characteristic | N (%) (Total *n* = 1468) | N (%) Depressed (*n* = 826) | *p* value[†] |
|---|---|---|---|
| *Where do you live* | | | 0.610 |
| Bangkok | 586 (39.9) | 336 (57.3) | |
| City other than Bangkok | 531 (36.2) | 297 (55.9) | |
| Regional center/town | 277 (18.9) | 148 (53.4) | |
| Rural or remote area | 74 (5.0) | 45 (60.8) | |
| *Age* | | | 0.018 |
| 15–17 | 181 (12.3) | 87 (48.1) | |
| 18–24 | 1287 (87.7) | 739 (57.4) | |
| *Sexual Orientation* | | | 0.214 |
| Gay | 1209 (82.3) | 668 (55.3) | |
| Bisexual | 217 (14.8) | 131 (60.4) | |
| Heterosexual/straight | 42 (2.9) | 27 (64.3) | |
| *Employment Status* | | | 0.283 |
| Full-time | 383 (26.1) | 203 (53.0) | |
| Part-time | 273 (18.6) | 153 (56.0) | |
| Not working | 812 (55.3) | 470 (57.9) | |
| *Education Enrollment Status* | | | 0.091 |
| Full-time student | 813 (55.4) | 445 (54.7) | |
| Part-time student | 270 (18.4) | 168 (62.2) | |
| Not a student | 385 (26.2) | 213 (55.3) | |
| *Highest Education Completed* | | | 0.649 |
| Secondary or Less | 864 (58.8) | 493 (57.1) | |
| Vocational | 211 (14.4) | 113 (53.6) | |
| Bachelor's or Higher | 393 (26.8) | 220 (56.0) | |
| *Income per month* | | | 0.273 |
| < $100 | 491 (33.4) | 272 (55.4) | |
| $101 - $150 | 248 (16.9) | 144 (58.1) | |
| $ 151–300 | 341 (23.2) | 205 (60.1) | |
| $ 301–450 | 205 (14.0) | 114 (55.6) | |
| $ 451–600 | 89 (6.1) | 46 (51.7) | |
| $ 601–900 | 48 (3.3) | 20 (41.7) | |
| > $900 | 46 (3.1) | 25 (54.4) | |
| *Importance of Religion* | | | <0.001 |
| Very important | 712 (48.5) | 363 (51.0) | |
| Somewhat important | 616 (42.0) | 375 (60.9) | |
| Not important | 140 (9.5) | 88 (62.9) | |
| **Social Group** | | | |
| *People in school/work* | | | 0.002 |
| All | 491 (33.4) | 249 (50.7) | |
| Not everyone | 977 (66.6) | 577 (59.1) | |
| *Friends outside school/work* | | | <0.001 |
| All | 413 (28.1) | 200 (48.4) | |
| Not everyone | 1055 (71.9) | 626 (59.3) | |
| *Teachers or Boss* | | | <0.001 |
| All | 440 (30.0) | 212 (48.2) | |
| Not everyone | 1028 (70.0) | 614 (59.7) | |
| *Family members* | | | 0.052 |

*(Continued)*

**Table 1.** (Continued)

| Characteristic | N (%) (Total $n$ = 1468) | N (%) Depressed ($n$ = 826) | $p$ value[†] |
|---|---|---|---|
| All | 549 (37.4) | 291 (53.0) | |
| Not everyone | 919 (62.6) | 535 (58.2) | |
| **Social Support** | | | <0.001 |
| Low | 644 (43.9) | 419 (65.1) | |
| Medium | 376 (25.6) | 199 (52.9) | |
| High | 448 (30.5) | 208 (46.4) | |
| **IPV Type** | | | 0.002 |
| None | 1056 (71.9) | 566 (53.6) | |
| Perpetrator only | 82 (5.6) | 45 (54.9) | |
| Victim only | 108 (7.4) | 75 (69.4) | |
| Both perpetrator and victim | 222 (15.1) | 140 (63.1) | |

[†] $p$ value for the $\chi^2$ test.

disclosed to all (Table 3). Amongst people with "high" social support in their lives, there was a similar percentage between those who indicated full disclosure to all and those who had not disclosed to everyone. Considering those with "high" social support, their disclosure to all was highest for "People in school/work" and conversely, those with "low" social support exhibited the lowest disclosure to all for "Friends outside school/work". There was a statistically significant association noted between sexual orientation disclosure and social support amongst each social network of interest ($p$<0.001). 30% of people continually disclosed to everyone among "People in school/work", "Friends outside school/work", and "Teachers or Boss". Patterns of disclosure to "All" did not generally differ based on an individual's experiences with IPV, however higher reports of IPV were always seen among those who did not tell everyone. "Family members" were the social group who received the highest full disclosure while "Friends outside school/work" received the lowest disclosure. The highest reports of IPV were among those who did not tell all their "Friends outside school/work" in comparison to the three others

**Table 2. Unadjusted and adjusted associations for individual study variables.**

| Study Variables | Unadjusted OR (95% CI) | Adjusted* OR (95% CI) |
|---|---|---|
| **Full Disclosure** | | |
| People in school/work | 0.71 (0.57, 0.89) | 0.72 (0.58, 0.91) |
| Friends outside school/work | 0.64 (0.51, 0.81) | 0.65 (0.51, 0.82) |
| Teachers or Boss | 0.63 (0.50, 0.79) | 0.63 (0.50, 0.80) |
| Family members | 0.81 (0.65, 1.00) | 0.84 (0.67, 1.05) |
| **Social Support** | | |
| Low | 1.00 | 1.00 |
| Medium | 0.60 (0.47, 0.78) | 0.61 (0.47, 0.79) |
| High | 0.47 (0.36, 0.60) | 0.49 (0.38, 0.63) |
| **IPV Type** | | |
| None | 1.00 | 1.00 |
| Perpetrator only | 1.05 (0.67, 1.65) | 0.94 (0.59, 1.49) |
| Victim only | 1.97 (1.28, 3.02) | 1.89 (1.22, 2.93) |
| Both perpetrator and victim | 1.48 (1.10, 1.99) | 1.44 (1.06, 1.95) |

* Adjusted for Table 1 variables

**Table 3. Associations of sexual orientation disclosure with social support and IPV (*n = 1468*).**

| Social Group | Social Support | | | | IPV Type | | | | |
|---|---|---|---|---|---|---|---|---|---|
| | Low | Medium | High | *p* value[†] | None | Perpetrator Only | Victim Only | Both | *p* value[†] |
| *People in school/work* | | | | <0.001 | | | | | 0.757 |
| All | 142 (22.0) | 120 (31.9) | 229 (51.1) | | 347 (32.9) | 26 (31.7) | 40 (37.0) | 78 (35.1) | |
| Not everyone | 502 (78.0) | 256 (68.1) | 219 (48.9) | | 709 (67.1) | 56 (68.3) | 68 (63.0) | 144 (64.9) | |
| *Friends outside school/work* | | | | <0.001 | | | | | 0.616 |
| All | 109 (16.9) | 94 (25.0) | 210 (46.9) | | 287 (27.2) | 24 (29.3) | 33 (30.6) | 69 (31.1) | |
| Not everyone | 535 (83.1) | 282 (75.0) | 238 (53.1) | | 769 (72.8) | 58 (70.7) | 75 (69.4) | 153 (68.9) | |
| *Teachers or Boss* | | | | <0.001 | | | | | 0.607 |
| All | 121 (18.8) | 111 (29.5) | 208 (46.4) | | 306 (29.0) | 26 (31.7) | 36 (33.3) | 72 (32.4) | |
| Not everyone | 523 (81.2) | 265 (70.5) | 240 (53.6) | | 750 (71.0) | 56 (68.3) | 72 (66.7) | 150 (67.6) | |
| *Family members* | | | | <0.001 | | | | | 0.516 |
| All | 196 (30.4) | 138 (36.7) | 215 (48.0) | | 385 (36.5) | 29 (35.4) | 44 (40.7) | 91 (41.0) | |
| Not everyone | 448 (69.6) | 238 (63.3) | 233 (52.0) | | 671 (63.5) | 53 (64.6) | 64 (59.3) | 131 (59.0) | |

[†] *p* value for the $\chi^2$ test.

groups. Across all social groups, there were no statistically significant associations noted between sexual orientation disclosure and IPV experiences (*p*>0.050).

Among each of the individual social groups, adjustment for sociodemographic characteristics, social support, and IPV type was conducted (Table 4). Within Table 4 are three models that adjusted for sociodemographic characteristics, and each model only differs based on the inclusion of a social group, social support levels, or IPV type. The OR for sexual orientation disclosure was raised after social support adjustment within each social group, compared to the unadjusted analyses. An elevated OR of 0.84 (95% CI 0.66–1.06) for "People in school/work" was reported, however there were significant changes following an adjustment for IPV experiences from the unadjusted analysis (OR 0.72; 95% CI 0.57–0.90). While there was no significant changes produced from IPV experiences being accounted for (OR 0.64; 95% CI 0.50–0.81), there was a significant increase in the OR for depression among "Friends outside school/work" to 0.76 (95% CI 0.59–0.97) with social support adjustment. There was an increased, significant association for depression risk (OR 0.73; 95% CI, 0.57–0.93) following social support adjustment, and this pattern was seen amongst "Family members" who received full disclosure (OR 0.91; 95% CI 0.73–1.15). We noted that adjustment for IPV showed no change in OR compared to unadjusted analyses. Regarding the fully adjusted model for all four social groups which included all study variables, the full disclosure OR was consistently comparable to the individual ORs focused on social support adjustment.

Fig 2 summarizes the complex intersection between the variables of disclosure and sexual identity, social support, IPV experiences, while simultaneously illustrating the distinct associations that each has with mental health. While improved mental health outcomes and elevated social support are positively associated with full disclosure to others, there does not appear to be an association with experiences of IPV. It appears that for Thai GBM, social support levels and IPV experiences do not have any bi-directional or uniliteral association with one another.

## Discussion/Conclusion

Our study provides strong supporting evidence among Thai GBM for independent associations between full sexual orientation disclosure, IPV experiences, and social support with depression as an emotional health outcome. Full disclosure of one's sexual orientation was

**Table 4. The adjusted associations between sociodemographic characteristics, social support, and IPV factors among different groups.**

| Study Variables | Model 1* | Model 2** | Model 3*** |
|---|---|---|---|
| *People in school/work* | | | |
| **Full Disclosure** | 0.84 (0.66, 1.06) | 0.72 (0.57, 0.90) | 0.83 (0.65, 1.05) |
| **Social Support** | | | |
| Low | 1.00 | - | 1.00 |
| Medium | 0.62 (0.48, 0.81) | - | 0.61 (0.47, 0.80) |
| High | 0.51 (0.39, 0.67) | - | 0.51 (0.39, 0.66) |
| **IPV Type** | | | |
| None | - | 1.00 | 1.00 |
| Perpetrator only | - | 0.93 (0.59, 1.49) | 1.02 (0.64, 1.64) |
| Victim only | - | 1.91 (1.23, 2.96) | 1.95 (1.25, 3.04) |
| Both perpetrator and victim | - | 1.45 (1.07, 1.97) | 1.46 (1.08, 1.99) |
| *Friends outside school/work* | | | |
| **Full Disclosure** | 0.76 (0.59, 0.97) | 0.64 (0.50, 0.81) | 0.75 (0.58, 0.96) |
| **Social Support** | | | |
| Low | 1.00 | - | 1.00 |
| Medium | 0.62 (0.48, 0.81) | - | 0.62 (0.47, 0.81) |
| High | 0.53 (0.41, 0.69) | - | 0.53 (0.40, 0.69) |
| **IPV Type** | | | |
| None | - | 1.00 | 1.00 |
| Perpetrator only | - | 0.95 (0.59, 1.52) | 1.03 (0.64, 1.65) |
| Victim only | - | 1.92 (1.24, 2.98) | 1.96 (1.26, 3.06) |
| Both perpetrator and victim | - | 1.47 (1.08, 1.99) | 1.47 (1.08, 2.00) |
| *Teachers or Boss* | | | |
| **Full Disclosure** | 0.73 (0.57, 0.93) | 0.63 (0.49, 0.79) | 0.72 (0.56, 0.92) |
| **Social Support** | | | |
| Low | 1.00 | - | 1.00 |
| Medium | 0.63 (0.48, 0.82) | - | 0.62 (0.48, 0.81) |
| High | 0.53 (0.41, 0.69) | - | 0.53 (0.41, 0.69) |
| **IPV Type** | | | |
| None | - | 1.00 | 1.00 |
| Perpetrator only | - | 0.95 (0.59, 1.52) | 1.03 (0.64, 1.65) |
| Victim only | - | 1.93 (1.24, 3.00) | 1.97 (1.26, 3.08) |
| Both perpetrator and victim | - | 1.46 (1.08, 1.99) | 1.47 (1.08, 2.00) |
| *Family members* | | | |
| **Full Disclosure** | 0.91 (0.73, 1.15) | 0.83 (0.66, 1.04) | 0.90 (0.72, 1.14) |
| **Social Support** | | | |
| Low | 1.00 | - | 1.00 |
| Medium | 0.61 (0.47, 0.80) | - | 0.61 (0.47, 0.79) |
| High | 0.49 (0.38, 0.64) | - | 0.49 (0.38, 0.64) |
| **IPV Type** | | | |
| None | - | 1.00 | 1.00 |
| Perpetrator only | - | 0.94 (0.59, 1.50) | 1.03 (0.64, 1.65) |
| Victim only | - | 1.90 (1.23, 2.95) | 1.95 (1.25, 3.04) |
| Both perpetrator and victim | - | 1.45 (1.07, 1.97) | 1.46 (1.08, 1.99) |

* Adjusted for Social Support and Table 1 variables

** Adjusted for IPV type and Table 1 variables

*** Adjusted for all variables

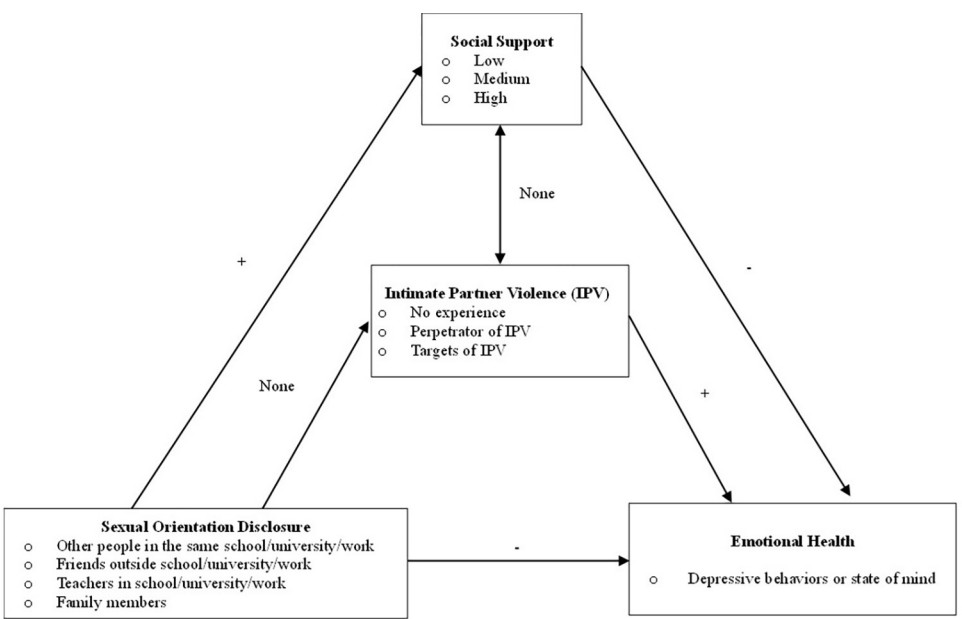

**Fig 2. The demonstrated associations between variables of full sexual orientation disclosure, IPV experiences, social support levels, and depression as an emotional health outcome.**

expected to cement larger, healthier connections to other people, and as expected, it was associated with greater social support. Elevated social support in itself was associated with lower odds of depression among respondents, thus lending support to our hypothesized mediation pathway. Although there were greater odds of depression associated with being an IPV victim, we found that IPV experiences did not vary by disclosure status, and thus was not a statistical mediator.

The results of this study illustrated a consistent and positive association between increased levels of disclosure and experiential levels of social support (Table 3). Gay, bisexual, and other men who have sex with men can be motivated to disclose their behavioral experiences and sexual orientation through a need for therapeutic resolve over stressors, including feelings of confusion or despair over being "closeted", while other men may be motivated to improve relationships and minimize emotional distance from others by sharing such information [11]. The trend of increased social support and disclosure was seen among all social groups, but not all GBM may feel comfortable disclosing to every community due to higher levels of sexual identity related stressors, and lacking a sense of comfort due to anticipated maltreatment within these networks. The potential negative responses that may ensue upon disclosure to others (i.e. bodily harm, social disapproval, social avoidance) can heavily influence whether to "come out" depending on the social context in which disclosure ensues [10]. Hence, disclosure processes are likely dependent upon one's social environment and that lower levels of disclosure in certain contexts may be adaptive and resilient [45, 46]. By increasing "outness", GBM can elevate self-acceptance, authentic living, and reduce distress, thus strengthening social networks through healthy and more expressive friendships [7, 10, 47, 48].

A lack of disclosure in this study was associated with increased depression across all social groups, potentially due to GBM anticipation of social discrimination and deciding against opening up to certain hostile social groups. Having to contend with the potential for negative physical and social responses from others upon disclosure can weigh as heavy stressors, and result in people remaining "closeted" for protection, however this can ultimately fester feelings

of self-consciousness, depression, and low self-esteem [2, 10, 12]. Past research among men who have sex with men has shown that social network connections can be instrumental in protecting against the effects of distal minority stressors on their psychological well-being, and lowered support can exacerbate depressive symptoms [49]. Among "Family members", the association between full disclosure and depression was not statistically significant ($p$ = 0.052), possibly explained by people gaining acceptance from other communities in a substantially enough manner that buffered against family-related struggles.

There was a prevalence of 65.1% for depression outcomes among the majority of respondents categorized with low social support. This prevalence exhibited a downward trend with increased levels of social support, a pattern that is consistent among other studies that applied the same social provisions scale to conceptualize and measure self-perceived community engagement. Elevated levels of social support in a person's life suggest that notable levels of social worth tend to trend with increased positive self-reliance and a greater overall quality of social connections [17, 50, 51]. Those that have disclosed to social groups and received positive and supportive responses have highlighted on the strengthened relationships they have gained, marked by an openness and comfortability with engaging in topics of sexuality [2]. It may be that stronger relationship ties are potentially inspired by the feeling that one's sexuality and same-sex relationships are valid, enabling others to be open and receive, at least partially, support from others. With a greater belief that one is connected to an understanding community accepting of gay and bisexual orientations, this can help GBM discover allies in their networks which may be protective against depression.

Our study did not find any significant difference between types of IPV experiences among respondents when analyzed by each social group and the extent to which GBM disclosed to them. Within the context of Thailand, recent increased digital communication and Internet accessibility has led to notable increases in cyberbullying, and digital harassment, with one study among 15–24-year-olds demonstrating around 50% have witnessed harassment/violence in online/offline manners in the past year [52]. Such witnesses have a correlation with being perpetrators of violence and further being cyclically involved in violent scenarios as victims themselves. With a prominent number involved in violence and considering that IPV experiences were not significantly associated with disclosure to any social groups in our study, this may inspire a fervent normalization of violence that is tolerated and more easily perpetrated against high risk minority groups such as GBM.

This is particularly alarming given that on average, four out of ten individuals in Southeast Asian countries still reject neighbors who are lesbian or gay [53]. While Thailand is found to be more accepting as a gay-friendly locale, especially in comparison to Indonesia and Malaysia, there is a need for culturally competent mental health mobilization as 50% of LGBTQ+ individuals report being bullied [53, 54]. Aggression still exists towards even when such Thai individuals seek health care to alleviate their mental health issues, with 36.5% report being stereotyped by health service providers and 24.6% being harassed or ridiculed, illustrating a need for increased mental health practitioner empathy and cultural competency towards their sexual orientation minority clientele [54].

A distinct association between IPV experiences and depression was shown by the OR for IPV perpetrators being practically 1.0, meaning that in comparison to those with no reported IPV experiences, there was no significantly higher odds of depression for the "perpetrator only" group. One study reported that men who experienced minority stressors of verbal or physical harassment diminished their level of disclosure to others and were more likely to report experiences of IPV, while another study found that sexual orientation related victimization was significantly and positively associated with psychological IPV perpetration [16, 55]. In this study, the experience of being an IPV victim–alone and when coupled with the

experience of being an IPV perpetrator–is associated with depression. It is suggested that the limited social connections to other family or friends and increased isolation or depression that individuals feel could predispose them to higher risk of violence. A previous study reported on similar results that, in comparison to female opposite-sex IPV victims, poor self-perceived health status was twice as likely among same-sex victims [56].

We conceptualized that full disclosure everyone in a social group would safeguard against IPV, as there would likely be a greater likelihood of disclosing to a surrounding community that shows acceptance and understanding. In environments that are high in autonomy support, described as interpersonal acceptance support for a person's true self-expression, researchers have found that gay and bisexual individuals are more likely to increase the degree to which they disclose to others [57]. Having autonomy support in social groups significantly moderated reports of negative feelings of anger, depression, and low-esteem issues [57]. A separate study reported that men whose social networks that contained higher amounts of sex partners or "closeted" gay friends were associated with higher IPV victimization and perpetration, in contrast to the significantly lower reports among those who had more gay friends [24]. While such levels of lowered sexual orientation disclosure may be related to fear of negative emotional and social responses from others, it was suggested that these men may also not have access to a wider LGBTQ+ inclusive community, diminished access to social support, or culturally appropriate services [12, 24]. It was anticipated that tolerance would inspire bonds that could be helpful against IPV risk factors, however, the results did not support this idea. This study did not find any statistically significant associations between different social groups and IPV history, illustrating that IPV involvement is not affected by the networks one discloses to. There are multiple triggers found to generally predispose couples to IPV, such as financial hardship or substance usage, however same-sex specific factors may be more comprehensive and encompass internalized homophobia, or discrimination based on racist attitudes [29, 58–60]. It is possible that such IPV risk factors may prove more instrumental in their contribution to the minority stressors that GBM experience, such as negative self-image and sexual identity discomfort, which may compound if left unresolved.

## Limitations and future directions

There are some limitations of note, the first being the study's cross-sectional design. There may exist a reverse causality between the variables of sexual orientation disclosure and social support, with increased levels of disclosure contributing to community connections or vice versa. However, given the cross-sectional study design, causality cannot be inferred at all. Longitudinal research to address this limitation is urgently needed. The second limitation is regarding the web-based survey distribution, as it narrowly excluded those without stable internet access. A potential GBM respondent lacking a reliable internet access would likely seek an alternate avenue separates from the social gay-network apps that this study was promoted through. Another limitation is related to the survey itself, as questions regarding sexual orientation disclosure contain vague wording that leaves room for subjective interpretation among participants who are self-reporting their behaviors. This is particularly important since people may have unique perceptions regarding their disclosure status among their social groups, and certain social settings may not prove as important toward their well-being as other groups are.

Despite these limitations, our study provides strengthened evidence that supportive social networks can have impacts on an individual's mental health. These findings also elucidate a greater understanding of how different IPV experiences, either as a perpetrator, a victim, or both, can shapes health outcomes of depression. There are unique challenges that GBM

communities face daily that affect their long-term health, even if they do live in a country that may be more accepting in comparison to others. Efforts to inform and positively support GBM individuals may reduce the community stigma and in turn support them in their decision to disclose their sexual orientation. Additionally, concerted efforts to improve attitudes and understanding of GBM communities can be used to provide better informed medical care in mental health counseling. Finally, improving mutual respect between communities with GBM individuals and reducing the perceived harm that people may feel surrounding their own IPV experiences can lead to better-informed health care and provide an opportunity to connect people who have experienced IPV. Through this discussion we can begin to provide resources and improved strategies that improve the lives of GBM individuals by improving their self-confidence and longevity.

## Acknowledgments

We thank all study participants for their willingness to participate, and all study staff for their support.

## Author Contributions

**Conceptualization:** Worawalan Waratworawan, Mayur M. Desai, Thomas E. Guadamuz.

**Data curation:** Yamol Kongjareon.

**Formal analysis:** Eduardo Encina, Yamol Kongjareon, Mayur M. Desai.

**Funding acquisition:** Thomas E. Guadamuz.

**Investigation:** Eduardo Encina, Worawalan Waratworawan, Thomas E. Guadamuz.

**Methodology:** Yamol Kongjareon, Thomas E. Guadamuz.

**Project administration:** Worawalan Waratworawan, Thomas E. Guadamuz.

**Supervision:** Worawalan Waratworawan, Yamol Kongjareon, Mayur M. Desai, Thomas E. Guadamuz.

**Validation:** Yamol Kongjareon.

**Visualization:** Yamol Kongjareon.

**Writing – original draft:** Eduardo Encina.

**Writing – review & editing:** Worawalan Waratworawan, Yamol Kongjareon, Mayur M. Desai, Thomas E. Guadamuz.

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
