## [Decision Letter · Decision Letter 0]

11 Sep 2022

PONE-D-22-15018Sexual Orientation Disclosure and Depression among Thai Men Who Have Sex with Men: Associations with Intimate Partner Violence and Social SupportPLOS ONE

Dear Dr. Guadamuz,

Thank you for submitting your manuscript to PLOS ONE. After careful consideration, we feel that it has merit but does not fully meet PLOS ONE’s publication criteria as it currently stands. Therefore, we invite you to submit a revised version of the manuscript that addresses the points raised during the review process.

We look forward to receiving your revised manuscript.

Kind regards,

Jianhong Zhou

Staff Editor

PLOS ONE

Journal Requirements:

“We thank all study participants for their willingness to participate, and all study staff for their support. Thomas E. Guadamuz, Worawalan Waratworawan, and Yamol Kongjareon were supported through NIMH grant R01 MH119015.”

“No, the funders had no role in study design, data collection and analysis, decision to publish, or preparation of the manuscript.”

“NO authors have competing interests”

6. We note you have included a table to which you do not refer in the text of your manuscript. Please ensure that you refer to Table 2 in your text; if accepted, production will need this reference to link the reader to the Table.

Additional Staff Editor Comments : In your Methods section, please provide additional information on how the sample size was determined and why sample size calculation was not carried out before sample collection.

Please include additional information regarding the survey or questionnaire used in the study and ensure that you have provided sufficient details that others could replicate the analyses. For instance, if you developed a questionnaire as part of this study and it is not under a copyright more restrictive than CC-BY, please include a copy, in both the original language and English, as Supporting Information.

Reviewers' comments:

Reviewer's Responses to Questions

**Comments to the Author**

1. Is the manuscript technically sound, and do the data support the conclusions?

Reviewer #1: Partly

2. Has the statistical analysis been performed appropriately and rigorously? 

Reviewer #1: Yes

3. Have the authors made all data underlying the findings in their manuscript fully available?

Reviewer #1: No

4. Is the manuscript presented in an intelligible fashion and written in standard English?

Reviewer #1: Yes

5. Review Comments to the Author

Reviewer #1: Strengths

There is a great need for research on the mental health of LGBTQ communities, particularly in non-Western societies. Furthermore, understanding the relationship of depression and intimate partner violence is of great significance for global public health. As such, this research is critically important. The purpose of the study has merit and has the potential to contribute to the field.

Areas of Weaknesses

The article being reviewed is an obvious contribution to the field. The study’s focus on depression, IPV, and sexual orientation disclosure among Thai men and is important and there is a dearth of research in this area. However, the manuscript as currently written is not suitable for publication. I would recommend accepting with several major revisions needed. Below are some tips to improve the quality of the manuscript for future publication.

Introduction

The introduction of the paper is very disjointed and does not underscore the rationale for the study. Specifically, it is unclear how sexual orientation disclosure relates to depression and IPV. The authors need to review the introduction to clearly tell the reader why they feel the relationship exist and its importance.

There are several places where the authors make claims that aren’t defended by either a citation or example of the existing research in the area (see lines 53, 57, 64, 65). Adding examples of the literature and citations for the claims would strengthen the arguments.

The aim of the study is to examine sexual orientation disclosure of gay and bisexual men however the authors use the term MSM to refer to the study population which is a behavioral term. Since the authors are referring to identity, the authors should use gay and bisexual not MSM which include people across sexual orientations. Furthermore, the authors should be consistent in language. Currently the authors use sexual minority, LGBTQ, and gay community to refer to the population. Please be specific and use consistent and correct language.

On line 66, the authors use the term “surrounding spaces” but it is unclear what they are referring to. Please clarify.

Methodology issue

The authors should clarify what the Greater Mekong sub-region is for readers and identify the countries part of this region.

The authors should identify the name of the study/project the data comes from.

It would be useful to the readers to have an example of one of the items parts of the IPV questions used in the study.

There is conflicting information about the age of participants. In one part of the paper, it states only persons over the age of 18 are included but in another it indicates people from 15-24. Please clarify who is included.

Analysis issues

The authors in the study state they had an initial sample of 1496 but only 1468 are included for analysis. The authors need to explain how they came up with this sample. Why were people excluded from the analysis?

Discussion

A large issue with this paper is that the authors do not address this critical issue of what might impact sexual orientation disclosure or non-disclosure. The authors suggest disclosure is a good thing, however this omits the reality of how it might be harmful to disclose one’s sexual orientation. Furthermore, the authors suggest disclosure reduces depression however the analysis doesn’t allow from this level of analysis. You cannot determine direction based on this study. It could be that individuals who are depressed are less likely to disclosure their sexual orientation or that in fact disclosure leads to reduction of depression. The authors should address this limitation.

On line 345 the authors incorrect use the term sexual orientation when they should be using gender identity.

The authors should avoid causality. They cannot establish causality given the study design.

6. PLOS authors have the option to publish the peer review history of their article (what does this mean?). If published, this will include your full peer review and any attached files.

Reviewer #1: **Yes: **Darren Whitfield

---

## [Author Response · Author response to Decision Letter 0]

11 Jan 2023

Reviewer #1: Strengths

There is a great need for research on the mental health of LGBTQ communities, particularly in non-Western societies. Furthermore, understanding the relationship of depression and intimate partner violence is of great significance for global public health. As such, this research is critically important. The purpose of the study has merit and has the potential to contribute to the field.

Areas of Weaknesses

The article being reviewed is an obvious contribution to the field. The study’s focus on depression, IPV, and sexual orientation disclosure among Thai men and is important and there is a dearth of research in this area. However, the manuscript as currently written is not suitable for publication. I would recommend accepting with several major revisions needed. Below are some tips to improve the quality of the manuscript for future publication.

Introduction

The introduction of the paper is very disjointed and does not underscore the rationale for the study. Specifically, it is unclear how sexual orientation disclosure relates to depression and IPV. The authors need to review the introduction to clearly tell the reader why they feel the relationship exist and its importance.

- Response: The introduction of the paper has been revised to properly reflect the relationships between disclosure, depression and IPV to reinforce the importance of this study (Introduction, lines 62-68, pages 3-4).

There are several places where the authors make claims that aren’t defended by either a citation or example of the existing research in the area (see lines 53, 57, 64, 65). Adding examples of the literature and citations for the claims would strengthen the arguments.

- Response: We have now included examples and provided proper citations (Introduction, lines 52-53, 56-57, 62-66, pages 3-4).

The aim of the study is to examine sexual orientation disclosure of gay and bisexual men however the authors use the term MSM to refer to the study population which is a behavioral term. Since the authors are referring to identity, the authors should use gay and bisexual not MSM which include people across sexual orientations. Furthermore, the authors should be consistent in language. Currently the authors use sexual minority, LGBTQ, and gay community to refer to the population. Please be specific and use consistent and correct language. On line 66, the authors use the term “surrounding spaces” but it is unclear what they are referring to. Please clarify.

- Response: We have now made the terms clearer and consistent throughout the manuscript. We feel that the term MSM is appropriate in the spaces where it is used because some participants identify as “straight” and not as “gay” or “bi.” 

- Additionally, “surrounding spaces” was meant to refer to research on surrounding spaces for gay men. It has now been edited for clarity (Introduction, lines 68-71, page 4).

Methodology issue

The authors should clarify what the Greater Mekong sub-region is for readers and identify the countries part of this region.

- Response: We have now clarified what the Greater Mekong sub-region is and the countries that are included in this region (Methods, lines 126-127, page 6).

The authors should identify the name of the study/project the data comes from.

- Response: This has been updated appropriately (Methods, lines 125-127, page 6). 

It would be useful to the readers to have an example of one of the items parts of the IPV questions used in the study.

- Response: Example of the IPV questions such as "Hurting, hitting, slapping the body of a regular partner, casual partner or male sex work partner", "Forcing a regular partner or casual partner to have sex", "Forcing a person who is a male sex work partner to have sex." and "Fondling or unwanted sexual touching" have now been included in the Methods section (Methods, lines 187-189, page 9). 

There is conflicting information about the age of participants. In one part of the paper, it states only persons over the age of 18 are included but in another it indicates people from 15-24. Please clarify who is included.

Analysis issues

- Response: We have checked and edited the manuscript so that all participants included in this study are age of 18 years and older. 

The authors in the study state they had an initial sample of 1496 but only 1468 are included for analysis. The authors need to explain how they came up with this sample. Why were people excluded from the analysis?

- Response: In our final analysis, we decided to only include those individuals who fully answered all survey questions and clarified this appropriately in the new draft (Results, lines 235-236, page 11).

Discussion

A large issue with this paper is that the authors do not address this critical issue of what might impact sexual orientation disclosure or non-disclosure. The authors suggest disclosure is a good thing, however this omits the reality of how it might be harmful to disclose one’s sexual orientation. Furthermore, the authors suggest disclosure reduces depression however the analysis doesn’t allow from this level of analysis. You cannot determine direction based on this study. It could be that individuals who are depressed are less likely to disclosure their sexual orientation or that in fact disclosure leads to reduction of depression. The authors should address this limitation.

- The reviewer is correct that this study is limited by being a cross-sectional study and hence cannot establish causal relationships among the variables. We have now revised our discussion to reflect this limitation. (Discussion, pages 19 – 22).

On line 345 the authors incorrect use the term sexual orientation when they should be using gender identity.

- Response: This sentence was removed and edited in the new updated manuscript (Discussion, pages 19 – 22).

---

## [Decision Letter · Decision Letter 1]

11 Apr 2023

PONE-D-22-15018R1Sexual Orientation Disclosure and Depression among Thai Men Who Have Sex with Men: Associations with Intimate Partner Violence and Social SupportPLOS ONE

Dear Dr. Guadamuz,

Thank you for submitting your manuscript to PLOS ONE. After careful consideration, we feel that it has merit but does not fully meet PLOS ONE’s publication criteria as it currently stands. Therefore, we invite you to submit a revised version of the manuscript that addresses the points raised during the review process.

We look forward to receiving your revised manuscript.

Kind regards,

Paolo Roma

Academic Editor

PLOS ONE

Additional Editor Comments:

Dear Authors,

as highlighted by both Reviewers, especially Reviewer 2, the manuscript still needs some work before it can be considered for publication.

I believe that the manuscript would strongly benefit from following Reviewers' insightful suggestions.

Reviewers' comments:

Reviewer's Responses to Questions

**Comments to the Author**

1. If the authors have adequately addressed your comments raised in a previous round of review and you feel that this manuscript is now acceptable for publication, you may indicate that here to bypass the “Comments to the Author” section, enter your conflict of interest statement in the “Confidential to Editor” section, and submit your "Accept" recommendation.

Reviewer #1: (No Response)

Reviewer #2: (No Response)

2. Is the manuscript technically sound, and do the data support the conclusions?

Reviewer #1: Partly

Reviewer #2: Partly

3. Has the statistical analysis been performed appropriately and rigorously? 

Reviewer #1: Yes

Reviewer #2: Yes

4. Have the authors made all data underlying the findings in their manuscript fully available?

Reviewer #1: No

Reviewer #2: No

5. Is the manuscript presented in an intelligible fashion and written in standard English?

Reviewer #1: Yes

Reviewer #2: Yes

6. Review Comments to the Author

Reviewer #1: I appreciate the author's attention to the feedback and responsiveness in their revision. I believe the article reads better overall. In general the authors have responded to all of my concerns. I still have a major concern which is about the sample. There is a conflation of sexual orientation and gender in the sample and analysis. The goal of the study is the examine sexual orientation disclosure and IPV. However the sample includes heterosexual men and they use the term MSM to define the sample. There is a clear incongruence here. Heterosexual men may be engaging in sexual behavior with other men but they are still not part of the LGBTQ community and it doesn't make sense to include them in the analysis because your question is related to identity not behavior. I would suggest omitting them or you need to change the angle of your paper and not focus on sexual orientation disclosure and LGBTQ issues.

Reviewer #2: Thank you for the opportunity to review this manuscript, which examines the associations between sexual orientation disclosure, IPV experiences, social support, and depressive symptoms among Thai gay and bisexual men. The revised version of the paper is generally well-written, and this is an important cross-cultural topic as a dearth of knowledge surrounds the nature, prevalence, and impact of minority stress in Thailand. Interesting associations are proposed. However, I have several suggestions, considerations, and questions that need to be addressed before considering the paper for publication.

Title

- Given the sample demographics, it might be more accurate and less misleading to use the term “gay, bisexual, and other men who have sex with men” throughout the manuscript or at least in the title instead of solely “MSM.”

Abstract

- “This article’s goal” should be reworded to sound less informal (e.g., “The purpose of this study,” “The goal of this study,” “The primary goal of this research,”)

- Should include that the majority of the sample identified as gay and bisexual. It cannot be assumed that the sample consists entirely of MSM participants unless you specify that in the Methods section of the abstract.

- The Discussion section of the abstract reads too much like the Results section. Rather than repeating findings, this section should synthesize and interpret major takeaways and highlight important implications. No implications are highlighted at the moment in this section.

Introduction

- The paper starts off by highlighting the experiences of LGBQ people specifically rather than MSM. I would recommend adding “questioning” to account for the small number of participants in your sample who may be questioning but still identify as heterosexual.

- Consider rewording the second sentence of the introduction to “Some key environmental and social influences that… include…”

- Make sure the terms you use throughout the manuscript when citing studies are consistent with their samples (e.g., LGBTQ+, LGB, LGBQ, MSM).

- Could add subheadings to improve the organization of the introduction.

- Although the proposed associations are explained in the introduction, there does not appear to be an explicitly stated theoretical framework that guides the conceptualized models. For example, you reference “stressors” and “social stressors” a couple of times throughout the manuscript and cite research guided by minority stress theory, but do not mention minority stress theory or any other established theories that might help bolster your hypotheses.

- There are conceptual nuances to your proposed model that should be addressed in the Introduction section and/or the Discussion. As demonstrated by your findings and previous findings that you cite, sexual orientation disclosure is largely associated with positive mental health outcomes, but not always, as sexual orientation disclosure can lead to more frequent instances of discrimination, which can harm mental health/well-being. It is possible that greater outness may result in lower social support and a greater reliance on partners for support due to discrimination, which may exacerbate IPV experiences. Therefore, it should be emphasized that these processes are likely highly contingent upon the individual’s social environment and that lower levels of disclosure in certain contexts may be adaptive, as evidenced in the following studies:

1) van der Star, A., Pachankis, J. E., & Bränström, R. (2021). Country-level structural stigma, school-based and adulthood victimization, and life satisfaction among sexual minority adults: A life course approach. Journal of Youth and Adolescence, 50(1), 189–201. https://doi.org/10.1007/s10964-020-01340-9

2) Shepherd, B. F., Chang, C. J., Dyar, C., Brochu, P. M., Selby, E. A., & Feinstein, B. A. (2022). Out of the closet, but not out of the woods: The longitudinal associations between identity disclosure, discrimination, and nonsuicidal self-injury among sexual minoritized young adults. Psychology of Sexual Orientation and Gender Diversity. Advance online publication. https://doi.org/10.1037/sgd0000597

- Lastly, the hypotheses of your study need to be more clearly stated in the Introduction section. I recommend creating a “Present Study” or “Current Study” subsection to briefly state the study’s overarching purpose and specific hypotheses, especially given the complexity of the proposed multiple mediation model.

Methods

- The Ns should be italicized in the tables and throughout the manuscript, as well as the p values.

- Based on this statement (“Attenuation of the effect of disclosure on depression with the addition of the potential mediators was taken as evidence supporting the hypothesized mediation pathway”) and the results (e.g., no “indirect” or “direct” effects/associations were stated), I believe the authors may be confusing moderation with mediation. Attenuation alone is not sufficient evidence to support a mediation pathway, especially since the data are cross-sectional, but social support and IPV experiences may both affect the strength or directionality of the association between sexual orientation disclosure and emotional health as potential moderators. Please clarify or use alternative methodologies to test statistical mediation such as Hayes’ (2017) PROCESS macro. Was mediation hypothesized, but not tested?

Results

- Subheadings are recommended to improve the organization of the Results section. There is a lot going on in this section and it is easy for readers to get lost, especially since a summary of the specific hypotheses tested was not provided in the Introduction section.

- I would avoid the use of the word “impact,” “led to,” and other words that may suggest causation between the variables of interest in the Results and Discussion sections, since the data are cross-sectional.

- In the tables, percentages of participants should be rounded so each column adds up to 100%.

Discussion

- The Discussion section has a lot of great information regarding sexual orientation disclosure, social support, IPV, and emotional health, but can be more strategically organized.

- The first paragraph is a summary of the study and your main findings, which is fine, but the following paragraphs read more like an Introduction section. In this section, each main finding should be explicitly stated, explained, and related to theories and prior research findings to highlight and support your contributions to the existing literature. You do this wonderfully in some paragraphs, but not so much in others.

- I would recommend adding subheadings to improve the organization of the Discussion section. For example, you could create a “Limitations and Future Directions” section.

- The authors state that there might be “reverse causality” because of the cross-sectional design of their study. However, because the data are cross-sectional, causation cannot be inferred at all, in any direction, and that needs to be clearly stated. I recommend that the authors encourage future longitudinal research to address this limitation.

Relatedly, the authors use too much causal/longitudinal language (e.g., “led to”) when discussing their findings, which again are based on cross-sectional data. Even the term mediator itself implies causation and should instead be called a “statistical mediator.” However, as previously stated, it is unclear whether the authors tested social support and IPV experiences as statistical mediators or jumped to conclusions based on independent associations/odds ratios.

Thank you again for the opportunity to review this manuscript, which touches on a very important topic and has a lot of potential.

7. PLOS authors have the option to publish the peer review history of their article (what does this mean?). If published, this will include your full peer review and any attached files.

Reviewer #1: **Yes: **Darren L Whitfield

Reviewer #2: No

---

## [Author Response · Author response to Decision Letter 1]

29 Jun 2023

Reviewer's Responses to Questions/Comments

Title

- Given the sample demographics, it might be more accurate and less misleading to use the term “gay, bisexual, and other men who have sex with men” throughout the manuscript or at least in the title instead of solely “MSM.” 

 - Response: We agreed and already changed the title. 

Abstract

- “This article’s goal” should be reworded to sound less informal (e.g., “The purpose of this study,” “The goal of this study,” “The primary goal of this research,”)

 - Response: We agreed and already reworded to “The primary goal of this research”.

- Should include that the majority of the sample identified as gay and bisexual. It cannot be assumed that the sample consists entirely of MSM participants unless you specify that in the Methods section of the abstract.

 - Response: We agreed and added gay, bisexual, and other men who have sex with men (GBM) to the methods section of the abstract. 

- The Discussion section of the abstract reads too much like the Results section. Rather than repeating findings, this section should synthesize and interpret major takeaways and highlight important implications. No implications are highlighted at the moment in this section.

 - Response: We agreed and have revised discussion on the abstract.

Introduction

- The paper starts off by highlighting the experiences of LGBQ people specifically rather than MSM. I would recommend adding “questioning” to account for the small number of participants in your sample who may be questioning but still identify as heterosexual.

 - Response: We agreed and have included “questioning” to our list of LGBTQ.

- Consider rewording the second sentence of the introduction to “Some key environmental and social influences that… include…”

 - Response: We have revised the sentence to “Some key environmental and social factors influence…”

- Make sure the terms you use throughout the manuscript when citing studies are consistent with their samples (e.g., LGBTQ+, LGB, LGBQ, MSM).

 - Response: We have now made sure the terms we use throughout the manuscript when citing studies are consistent with their samples.

- Could add subheadings to improve the organization of the introduction.

 - Response: We have now added subheadings.

- Although the proposed associations are explained in the introduction, there does not appear to be an explicitly stated theoretical framework that guides the conceptualized models. For example, you reference “stressors” and “social stressors” a couple of times throughout the manuscript and cite research guided by minority stress theory, but do not mention minority stress theory or any other established theories that might help bolster your hypotheses.

 - Response: We have now re-structured to the Introduction to include a discussion on the minority stress theory and how it relates to the current study. 

- There are conceptual nuances to your proposed model that should be addressed in the Introduction section and/or the Discussion. As demonstrated by your findings and previous findings that you cite, sexual orientation disclosure is largely associated with positive mental health outcomes, but not always, as sexual orientation disclosure can lead to more frequent instances of discrimination, which can harm mental health/well-being. It is possible that greater outness may result in lower social support and a greater reliance on partners for support due to discrimination, which may exacerbate IPV experiences. Therefore, it should be emphasized that these processes are likely highly contingent upon the individual’s social environment and that lower levels of disclosure in certain contexts may be adaptive, as evidenced in the following studies:

1) van der Star, A., Pachankis, J. E., & Bränström, R. (2021). Country-level structural stigma, school-based and adulthood victimization, and life satisfaction among sexual minority adults: A life course approach. Journal of Youth and Adolescence, 50(1), 189–201. https://doi.org/10.1007/s10964-020-01340-9

2) Shepherd, B. F., Chang, C. J., Dyar, C., Brochu, P. M., Selby, E. A., & Feinstein, B. A. (2022). Out of the closet, but not out of the woods: The longitudinal associations between identity disclosure, discrimination, and nonsuicidal self-injury among sexual minoritized young adults. Psychology of Sexual Orientation and Gender Diversity. Advance online publication. https://doi.org/10.1037/sgd0000597

 - Response: We have now emphasized that these processes are likely highly contingent upon the individual’s social environment and that lower levels of disclosure in certain contexts may be adaptive. We have also included two studies you kindly provided for us. 

- Lastly, the hypotheses of your study need to be more clearly stated in the Introduction section. I recommend creating a “Present Study” or “Current Study” subsection to briefly state the study’s overarching purpose and specific hypotheses, especially given the complexity of the proposed multiple mediation model.

 - Response: We have included “Current Study” subsection to briefly state the study’s overarching purpose and specific hypotheses.

Methods

- The Ns should be italicized in the tables and throughout the manuscript, as well as the p values.

 - Response: Ns have now been italicized in the tables and throughout the manuscript, as well as the p values.

- Based on this statement (“Attenuation of the effect of disclosure on depression with the addition of the potential mediators was taken as evidence supporting the hypothesized mediation pathway”) and the results (e.g., no “indirect” or “direct” effects/associations were stated), I believe the authors may be confusing moderation with mediation. Attenuation alone is not sufficient evidence to support a mediation pathway, especially since the data are cross-sectional, but social support and IPV experiences may both affect the strength or directionality of the association between sexual orientation disclosure and emotional health as potential moderators. Please clarify or use alternative methodologies to test statistical mediation such as Hayes’ (2017) PROCESS macro. Was mediation hypothesized, but not tested?

 - Response: We have improved the methods section by including a clarification of the statistical mediators of interest.

Results

- Subheadings are recommended to improve the organization of the Results section. There is a lot going on in this section and it is easy for readers to get lost, especially since a summary of the specific hypotheses tested was not provided in the Introduction section.

 - Response: We have now added subheadings in the results section.

- I would avoid the use of the word “impact,” “led to,” and other words that may suggest causation between the variables of interest in the Results and Discussion sections, since the data are cross-sectional.

 - Response: We have now replaced “impact” with “association” throughout the manuscript. 

- In the tables, percentages of participants should be rounded so each column adds up to 100%.

 - Response: We have now rounded up percentages of participants so that each column adds up to 100%.

Discussion

- The Discussion section has a lot of great information regarding sexual orientation disclosure, social support, IPV, and emotional health, but can be more strategically organized.

 - Response: We have now reorganized the discussion section as suggest.

- The first paragraph is a summary of the study and your main findings, which is fine, but the following paragraphs read more like an Introduction section. In this section, each main finding should be explicitly stated, explained, and related to theories and prior research findings to highlight and support your contributions to the existing literature. You do this wonderfully in some paragraphs, but not so much in others.

 - Response: We agreed and have now made sure that the discussion section does not read like the introduction section and each main finding is explicitly stated, explained, and related to theories and prior research findings to highlight and support the study’s contributions to the existing literature.

- I would recommend adding subheadings to improve the organization of the Discussion section. For example, you could create a “Limitations and Future Directions” section.

 - Response: We have now added subheading to improve the organization of the Discussion section.

- The authors state that there might be “reverse causality” because of the cross-sectional design of their study. However, because the data are cross-sectional, causation cannot be inferred at all, in any direction, and that needs to be clearly stated. I recommend that the authors encourage future longitudinal research to address this limitation.

 - Response: We have included a statement to emphasize that causality cannot be inferred at all since this is a cross-sectional study. Longitudinal research to address this limitation urgently needed.

Relatedly, the authors use too much causal/longitudinal language (e.g., “led to”) when discussing their findings, which again are based on cross-sectional data. Even the term mediator itself implies causation and should instead be called a “statistical mediator.” However, as previously stated, it is unclear whether the authors tested social support and IPV experiences as statistical mediators or jumped to conclusions based on independent associations/odds ratios.

 - Response: We have replaced the word mediators with statistical mediators throughout the manuscript. They are social support and IPV experiences.

---

## [Decision Letter · Decision Letter 2]

1 Aug 2023

PONE-D-22-15018R2Sexual orientation disclosure and depression among gay, bisexual, and other men who have sex with men: Associations with intimate partner violence and social supportPLOS ONE

Dear Dr. Guadamuz,

Thank you for submitting your manuscript to PLOS ONE. After careful consideration, we feel that it has merit but does not fully meet PLOS ONE’s publication criteria as it currently stands. Therefore, we invite you to submit a revised version of the manuscript that addresses the points raised during the review process.

Please submit your revised manuscript by Sep 15 2023 11:59PM. If you will need more time than this to complete your revisions, please reply to this message or contact the journal office at plosone@plos.org. Please include the following items when submitting your revised manuscript:A rebuttal letter that responds to each point raised by the academic editor and reviewer(s). You should upload this letter as a separate file labeled 'Response to Reviewers'.A marked-up copy of your manuscript that highlights changes made to the original version. You should upload this as a separate file labeled 'Revised Manuscript with Track Changes'.An unmarked version of your revised paper without tracked changes. You should upload this as a separate file labeled 'Manuscript'.If applicable, we recommend that you deposit your laboratory protocols in protocols.io to enhance the reproducibility of your results. Protocols.io assigns your protocol its own identifier (DOI) so that it can be cited independently in the future. For instructions see: https://journals.plos.org/plosone/s/submission-guidelines#loc-laboratory-protocols. Additionally, PLOS ONE offers an option for publishing peer-reviewed Lab Protocol articles, which describe protocols hosted on protocols.io. Read more information on sharing protocols at https://plos.org/protocols?utm_medium=editorial-email&utm_source=authorletters&utm_campaign=protocols.

We look forward to receiving your revised manuscript.

Kind regards,

Paolo Roma

Academic Editor

PLOS ONE

Journal Requirements:

Reviewers' comments:

Reviewer's Responses to Questions

**Comments to the Author**

1. If the authors have adequately addressed your comments raised in a previous round of review and you feel that this manuscript is now acceptable for publication, you may indicate that here to bypass the “Comments to the Author” section, enter your conflict of interest statement in the “Confidential to Editor” section, and submit your "Accept" recommendation.

Reviewer #1: (No Response)

Reviewer #2: (No Response)

2. Is the manuscript technically sound, and do the data support the conclusions?

Reviewer #1: Yes

Reviewer #2: Yes

3. Has the statistical analysis been performed appropriately and rigorously? 

Reviewer #1: Yes

Reviewer #2: Yes

4. Have the authors made all data underlying the findings in their manuscript fully available?

Reviewer #1: Yes

Reviewer #2: No

5. Is the manuscript presented in an intelligible fashion and written in standard English?

Reviewer #1: Yes

Reviewer #2: Yes

6. Review Comments to the Author

Reviewer #1: I appreciate the author's consideration of the reviewer's comments. I believe the current iteration is improved. I only have minor suggestions to improve the article.

I would suggest the title indicate Mekong Region.

The introduction is greatly approved. I would suggest the authors provide either a theoretical or empirical rationale for why social support is included.

Reviewer #2: I'm quite satisfied with the revisions that have been made based on the rounds of review and look forward to seeing the article published in PLOS One. My only remaining concern is the future directions section at the end of the paper. Research limitations are adequately described, but the real life applications of your paper are not. For example, you briefly mention implications for public policy and community efforts/resources in the abstract and introduction, but do not clearly tie your findings to these implications in the discussion section even though implications for policy and community resources are critical to the purpose and theoretical basis for your paper (e.g., structural/community stigma leads to increased sexual orientation concealment and decreased community resources). Implications for clinical practice (e.g., affirmative mental health conceptualizations/interventions) could also be described. Overall, your paper aims to increase awareness on ways to mitigate depressive mental health outcomes, but needs to provide readers (especially policy makers, community advocates, and mental health practicioners) with a better understanding of how this awareness can be translated into action.

I also think the concluding sentence of the paper can improved upon/reframed to end on a stronger note. Too many ideas are crammed into a single sentence and overgeneralizations are used. I would avoid calling the entire country "extremely tolerant and gay-friendly." If this was the case, sexual orientation disclosure would not be an issue. The word tolerant also implies that sexual diversity is something to be "tolerated" rather than accepted or embraced, so I would avoid using that language as well in academic writing.

Recommended readings include:

-Layland, E.K., Bränström, R., Murchison, G.R. et al. Kept in the Closet: Structural Stigma and the Timing of Sexual Minority Developmental Milestones Across 28 European Countries. J Youth Adolescence (2023). https://doi.org/10.1007/s10964-023-01818-2

-Pachankis, J. E., Soulliard, Z. A., Morris, F., & Seager van Dyk, I. (2023). A model for adapting evidence-based interventions to be LGBQ-affirmative: Putting minority stress principles and case conceptualization into clinical research and practice. Cognitive and Behavioral Practice, 30(1), 1–17. https://doi.org/10.1016/j.cbpra.2021.11.005

7. PLOS authors have the option to publish the peer review history of their article (what does this mean?). If published, this will include your full peer review and any attached files.

Reviewer #1: No

Reviewer #2: No

---

## [Author Response · Author response to Decision Letter 2]

14 Oct 2023

Responses to Reviewers’ Questions/Comments

Sexual Orientation Disclosure and Depression among Thai Men Who Have Sex with Men: The roles of Social Support and Intimate Partner Violence

Dear editor and reviewers: We thank the editor and reviewers for your feedback, suggestions and the recommended readings. We have subsequently edited the paper according to the suggestions which we feel have greatly improved the quality of the paper. We have included a discussion on the importance of Social Support as an important variable in this paper in the Introduction and have adjusted the title. We also updated and adjusted the future directions and conclusion of the paper. Finally, we have edited the paper for better flow and had a native English speaker make final edits throughout the paper. Please find below a point-by-point response to the reviewers’ comments. 

Reviewer #1: I appreciate the author's consideration of the reviewer's comments. I believe the current iteration is improved. I only have minor suggestions to improve the article.

1.1 I would suggest the title indicate Mekong Region.

Response: While the name of the study is “Greater Mekong - Young MSM Internet Survey”, only the Thai data was analyzed for this paper. We have now added the word “Thai” in the title to be clearer to the readers.

1.2 The introduction is greatly improved. I would suggest the authors provide either a theoretical or empirical rationale for why social support is included.

Response: We have now added to the Introduction an empirical rationale for why social support is included in the analysis.

Reviewer #2: I'm quite satisfied with the revisions that have been made based on the rounds of review and look forward to seeing the article published in PLOS One. My only remaining concern is the future directions section at the end of the paper. 

2.1 Research limitations are adequately described, but the real-life applications of your paper are not. For example, you briefly mention implications for public policy and community efforts/resources in the abstract and introduction, but do not clearly tie your findings to these implications in the discussion section even though implications for policy and community resources are critical to the purpose and theoretical basis for your paper (e.g., structural/community stigma leads to increased sexual orientation concealment and decreased community resources). 

Response: We have now added some real-life applications of the findings of our paper to the future directions section at the end of the paper. 

2.2 Implications for clinical practice (e.g., affirmative mental health conceptualizations/interventions) could also be described. Overall, your paper aims to increase awareness on ways to mitigate depressive mental health outcomes, but needs to provide readers (especially policy makers, community advocates, and mental health practitioners) with a better understanding of how this awareness can be translated into action.

Response: We have now added implications to the discussion section of the paper. 

2.3 I also think the concluding sentence of the paper can be improved upon/reframed to end on a stronger note. Too many ideas are crammed into a single sentence and overgeneralizations are used. I would avoid calling the entire country "extremely tolerant and gay-friendly." If this was the case, sexual orientation disclosure would not be an issue. The word tolerant also implies that sexual diversity is something to be "tolerated" rather than accepted or embraced, so I would avoid using that language as well in academic writing.

Response: We have now reframed the concluding sentence of the paper.

---

## [Decision Letter · Decision Letter 3]

3 Nov 2023

Sexual orientation disclosure and depression among Thai gay, bisexual, and other men who have sex with men: The role of social support and intimate partner violence

PONE-D-22-15018R3

Dear Dr. Guadamuz,

We’re pleased to inform you that your manuscript has been judged scientifically suitable for publication and will be formally accepted for publication once it meets all outstanding technical requirements.

Kind regards,

Paolo Roma

Academic Editor

PLOS ONE

Additional Editor Comments (optional):

Reviewers' comments:

Reviewer's Responses to Questions

**Comments to the Author**

1. If the authors have adequately addressed your comments raised in a previous round of review and you feel that this manuscript is now acceptable for publication, you may indicate that here to bypass the “Comments to the Author” section, enter your conflict of interest statement in the “Confidential to Editor” section, and submit your "Accept" recommendation.

Reviewer #1: All comments have been addressed

Reviewer #2: All comments have been addressed

2. Is the manuscript technically sound, and do the data support the conclusions?

Reviewer #1: Yes

Reviewer #2: Yes

3. Has the statistical analysis been performed appropriately and rigorously? 

Reviewer #1: Yes

Reviewer #2: Yes

4. Have the authors made all data underlying the findings in their manuscript fully available?

Reviewer #1: No

Reviewer #2: No

5. Is the manuscript presented in an intelligible fashion and written in standard English?

Reviewer #1: Yes

Reviewer #2: Yes

6. Review Comments to the Author

Reviewer #1: Thank you for your responsiveness to the feedback. I have no further comments. I look forward to seeing this paper in print.

Reviewer #2: I'm quite satisfied with the revisions that have been made based on the multiple rounds of review and look forward to seeing the article published in PLOS ONE.

7. PLOS authors have the option to publish the peer review history of their article (what does this mean?). If published, this will include your full peer review and any attached files.

Reviewer #1: No

Reviewer #2: No

---

## [Editor Report · Acceptance letter]

13 Nov 2023

PONE-D-22-15018R3 

Sexual orientation disclosure and depression among Thai gay, bisexual, and other men who have sex with men: The roles of social support and intimate partner violence 

Dear Dr. Guadamuz:

I'm pleased to inform you that your manuscript has been deemed suitable for publication in PLOS ONE. Congratulations! Your manuscript is now with our production department. 

Kind regards, 

on behalf of

Prof. Paolo Roma 

Academic Editor

PLOS ONE